# Dual-Criterion Quality Loss for Blind Image Quality Assessment

Desen Yuan*
desenyuan@gmail.com
University of Electronic Science and Technology of China
Chengdu, Sichuan, China

Lei Wang*
lwang@std.uestc.edu.cn
University of Electronic Science and Technology of China
Chengdu, Sichuan, China

## Abstract

This paper introduces a novel approach to Image Quality Assessment (IQA) by presenting a new loss function, Dual-Criterion Quality (DCQ) Loss, which integrates the Mean Squared Error (MSE) framework with a Relative Perception Constraint (RPC). The RPC is comprised of two main components: the Quantitative Discrepancy Constraint (QDC) and the Qualitative Alignment Constraint (QAC). The QDC focuses on capturing the numerical relationships of relative differences by minimizing the mean squared error between the differences in predicted scores among samples within a batch size and the differences in Mean Opinion Scores (MOS). Meanwhile, the QAC aims to capture the ordinal relationships between these differences. This method is designed to closely align with human subjective assessments of image quality, which are frequently quantified using the MOS, and to enhance the interpretability and reliability of IQA. Unlike existing ranking methods that suffer from complex pipelines and the introduction of errors through the generation of pair-wise or ordering data, DCQ Loss provides a more straightforward and efficient approach. Moreover, the loss function outperforms current rank-based IQA methods in terms of convergence, stability, and the ability to emulate human perception of visual quality. The effectiveness of this approach is validated through extensive experiments on various mainstream datasets and IQA network architectures, demonstrating significant performance gains over traditional rank loss approaches and contributing to the ongoing development of IQA.

## CCS Concepts

• **Computing methodologies → Image processing; Neural networks;**.

## Keywords

Image Quality Assessment; Blind Image Quality Assessment; No-Reference;

**ACM Reference Format:**
Desen Yuan and Lei Wang. 2024. Dual-Criterion Quality Loss for Blind Image Quality Assessment. In *Proceedings of the 32nd ACM International Conference on Multimedia (MM '24), October 28-November 1, 2024, Melbourne,*

---

*Both authors contributed equally to this research.

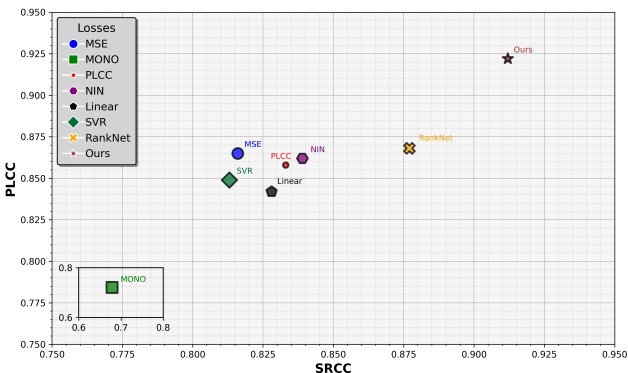

**Figure 1: SRCC v.s PLCC of different losses for DBCNN on TID2013 dataset. To make the figure appear more compact, the lower left corner of the diagram is designated as the second coordinate area, which contains the results for MONO.**

*VIC, Australia.* ACM, New York, NY, USA, 10 pages. https://doi.org/10.1145/3664647.3681250

## 1 Introduction

Images serve as a crucial form of data across a multitude of sectors. However, these images are often subject to quality degradation due to various factors such as acquisition methods, transmission processes, and storage conditions. Image Quality Assessment (IQA) [4, 11, 14, 22, 34, 64, 73] is an automated technique designed to gauge the perceptual quality of images. One of the key objectives of IQA is to generate estimates that align closely with human subjective assessments, which are frequently quantified using the Mean Opinion Score (MOS). Blind IQA [20, 20, 24, 39, 63, 68] has emerged as a focal point in IQA research, offering the advantage of assessing quality without the necessity for a reference image. IQA is especially crucial in specialized sectors that require rigorous image fidelity[41–43], such as medical imaging, video streaming services.

Certainly, the essence of IQA [3, 13, 38, 44–47, 54, 60, 71, 73] lies in a ranking task designed to evaluate and order images based on their perceptual quality. Strangely enough, this fundamental characteristic often seems overlooked in contemporary IQA research. Despite the apparent shift of focus, it's crucial to recognize that the domain of ranking-based IQA is far from being exhausted. In fact, there is substantial room for further research, especially in leveraging advanced machine learning algorithms to emulate human perception more accurately. Various scholars have made notable contributions in this area, such as Ma et al.'s dipIQ [23] and Liu et al.'s RankIQA [21], both seminal works in the application of learning-to-rank methodologies for IQA.

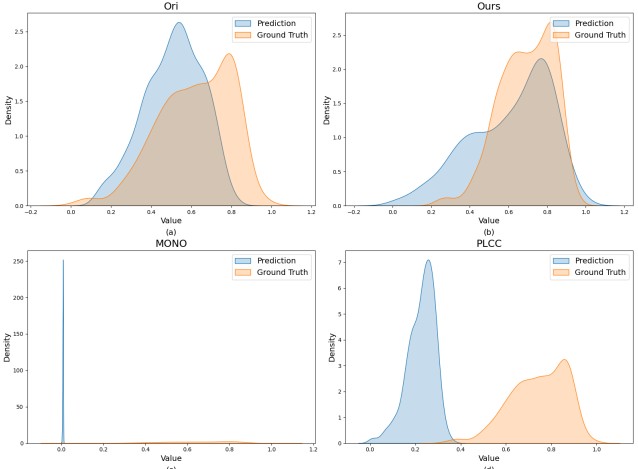

**Figure 2: The density plot results of the predicted scores and Ground Truth for four types of loss functions after normalization (TID2013, DBCNN) show that the density distributions of our loss function and the MSE loss function largely overlap. However, the density distributions of PLCC and MONO differ significantly, especially for MONO, where the predicted distribution is comparatively narrower.**

Existing ranking methods in IQA face challenges due to their complex pipelines and the need for generating pair-wise or ordering data, which complicates their integration into IQA frameworks and can introduce errors. This complexity signals a critical demand for simpler, yet efficient, ranking techniques. In the field of IQA, mainstream deep neural networks typically utilize Mean Squared Error (MSE) loss for optimization, focusing on minimizing the discrepancy between predicted quality scores and Mean Opinion Scores (MOS). Additionally, some research methods employ loss functions with ranking characteristics to aid image quality evaluation models in better learning the relationships between image qualities. As shown in Fig. 1, the corresponding loss function on the TID2013 dataset, utilizing DBCNN, produced results in Spearman's Rank Correlation Coefficient (SRCC) and Pearson's Linear Correlation Coefficient (PLCC).

Li et al. [19] introduced a monotonicity loss (MONO for short) to learn the monotonicity between scores. However, the ReLU truncation of its function limits the differentiation between correctly predicted images, resulting in overly similar output scores and poor generalizability. Compared to MSE, monotonicity loss is less conducive to learning the numerical relationship between predictions and Ground Truth. Li et al. [18] proposed a NIN loss aimed at enhancing convergence and performance, but this also leads to higher computational complexity due to numerous conditional judgments, its unique form being PLCC. Wu et al. [48, 49] used PLCC as a loss function. Although PLCC directly corresponds to the objective of image quality assessment as a metric, its optimization efficiency, handling of non-linearities, and generalization capabilities are inferior to those of MSE.

Although Mean Squared Error is suitable for regression problems related to image quality assessment, it overlooks a critical aspect of

human perception: the relative ranking of image quality. To address this shortfall while maintaining compatibility with existing Image Quality Assessment frameworks, we introduce a novel loss function called Dual-Criterion Quality (DCQ) Loss. DCQ Loss is derived from the original MSE loss, augmented with a ranking constraint that we term Relative Perception Constraint (RPC). The concept of relative perceptual constraints consists of two components: the Quantitative Discrepancy Constraint (QDC) and the Qualitative Alignment Constraint (QAC). The Quantitative Discrepancy Constraint is utilized to capture the numerical relationships of relative differences. It achieves this by minimizing the mean squared error between the differences in predicted scores among samples within a batch size and the differences in Mean Opinion Scores. The Qualitative Alignment Constraint is employed to capture the ordinal relationships between relative differences.

This DCQ Loss is crucial for ensuring that the model's predictions resonate with the human ability to recognize and rank visual quality, thereby enhancing the interpretability and reliability of image quality assessments. Compared to other ranking losses, such as monotonicity loss and PLCC loss, predictions are closer to the MOS, the predicted score range is not overly dense, the distribution is more uniform, and the training is more stable. As shown in Fig. 2, density plot results for four types of loss functions were selected, including Mean Squared Error, the loss function we proposed, MONO, and PLCC. The density plot results demonstrate that the predictive distribution of the loss function we proposed and MSE are closest to the Ground Truth. In contrast, the distribution overlap for MONO and PLCC is poor, with MONO exhibiting a very narrow prediction range.

Our contributions are as follows:

- We introduce the Dual-Criterion Quality Loss, which enhances image quality prediction by integrating numerical and ordinal relationship constraints.
- Experimental and theoretical analysis have confirmed the effectiveness of our method, which significantly outperforms existing approaches.

## 2 Related Work

### 2.1 Blind Image Quality Assessment

The realm of Blind/No Reference Image Quality Assessment has evolved from traditional methods employing hand-engineered features like Natural Scene Statistics (NSS) [7, 9, 27–30, 35, 53, 66] to modern, deep learning-oriented approaches [23, 56–58, 65, 67, 70, 72]. These initial methods mainly excelled on synthetically distorted images but faltered with real-world distortions. The infusion of deep learning has fostered end-to-end optimization, intertwining feature extraction and quality regression, thereby advancing BIQA/NRIQA significantly. This shift instigated exploration into diverse computational structures like generalized divisive normalization [25], adaptive convolution [38], and objective functions such as $\ell_p$-norm induced metric [40].

Deep learning in BIQA/NRIQA [2–4, 13, 14, 20, 20, 22, 24, 33, 34, 38, 39, 54, 63, 68, 71, 73] has leveraged Convolutional Neural Networks (CNNs), often pretrained on extensive datasets like ImageNet, to develop nuanced approaches. For instance, multi-task networks

[24] were devised for concurrent distortion identification and quality prediction. Additionally, innovative frameworks emerged, utilizing Generative Adversarial Models for hallucinated reference image generation [20], and meta-learning [73] to harness shared prior knowledge among different distortion types. The advent of novel frameworks like Vision-Transformers [59] and Cascaded CNNs [50] further embodies the dynamic nature of BIQA/NRIQA, promising a trajectory towards more robust and precise image quality assessment, marking a notable departure from conventional methodologies.

## 2.2 Learning to Rank

IQA (Image Quality Assessment) is widely regarded as a ranking problem [6, 21, 24]. Zhang et al. [68] utilized discrete ranking information derived from images with identical content but varying levels of distortion for quality prediction. Subsequently, Zhang et al. [68, 70] employed continuous ranking information derived from MOSs (Mean Opinion Scores) and the differences between subjective scores. Ma et al. [24, 26] extracted binary ranking information during the training process through NR-IQA methods. However, due to the employment of reference images, their method is only applicable to synthetic distortions.

dipIQ [23] and RankIQA [21] have employed siamese networks with generated data pairs for ranking learning. However, such approaches, involving paired dataset generation or dual networks, are complex and less suited for Blind Image Quality Assessment (BIQA) due to their complexity and lack of convenience. Wu et al. [52] proposed a regression model for BIQA using rank regularization, but this model does not consider the specific structures or discrepancies between predicted scores and MOSs, and minimizing mean error alone may not ensure accurate quality ranking. Alireza et al. [10] developed a network that integrates feature fusion with Transformers and an attention mechanism, along with relative ranking and consistency representation. However, this method is computationally complex and dependent on extreme samples and margin settings. Li et al. [18] introduced a NIN loss function to improve BIQA loss, aiming to enhance convergence and performance, but this also leads to high computational complexity due to numerous conditional judgments. These methods overlook the inherent rank information in the original samples.

Moreover, ranking learning, originating from information retrieval, includes pointwise, pairwise, and listwise concepts with representative loss functions like RankNet [5], used in BIQA research. Our proposed DCQ Loss method stands out in this context for its generality, low computational demand, and ease of integration into various BIQA frameworks, effectively learning the pointwise, pairwise, and listwise ranking information without needing data synthesis.

## 3 Dual-Criterion Quality Loss

The Dual-Criterion Quality Loss aims to address the lack of information on image ranking inherent in the Mean Squared Error loss function, as well as the current inaccuracies in the prediction range of Mean Opinion Score by ranking loss. Dual-Criterion Quality Loss offers a stable, highly effective, and concise form of loss function.

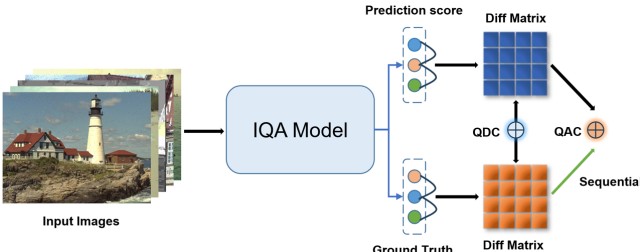

**Figure 3: The proposed loss function is integrated with the architecture of the IQA model. QDC calculates the MSE of two difference matrices, whereas, in the calculation of QAC, the difference matrix of the Ground Truth only provides information about the Sequential.**

## 3.1 Loss Definition

Given a batch of input images $x$ and their corresponding ground truth Mean Opinion Scores $t$, let $\hat{y} = \text{model}(x)$ denote the predicted quality scores by the model. The Relative Perception Constraint $\mathcal{L}$, combining QDC and QAC, is defined as:

$$\mathcal{L} = \frac{1}{N^2} \sum_{i=1}^{N} \sum_{j=1}^{N} \left[ \left( \Delta\hat{y}_{ij} - \Delta t_{ij} \right)^2 - \Delta\hat{y}_{ij} \cdot \text{sgn}(\Delta t_{ij}) \right], \quad (1)$$

where $\Delta\hat{y}_{ij} = \hat{y}_i - \hat{y}_j$ represents the difference between predicted scores, and $\Delta t_{ij} = t_i - t_j$ represents the difference between the Mean Opinion Scores. The function $\text{sgn}(\cdot)$ denotes the sign function, which indicates the direction of the difference. Combining RPC with the MSE of prediction and MOS results in the proposed DCQ Loss.

The first term inside the summation, $(\hat{y}_i - \hat{y}_j - (t_i - t_j))^2$, represents the Mean Squared Error between the differences in predicted scores and actual MOS values for all pairs of images within a batch. This term penalizes deviations in the magnitude of score differences, encouraging precise estimation of the score gaps as per the ground truth.

The second term, $-(\hat{y}_i - \hat{y}_j) \cdot \text{sign}(t_i - t_j)$, introduces a ranking loss that penalizes incorrect relative ordering of the predicted scores with respect to the true MOS. By incorporating the sign of the ground truth differences, this term ensures that the model is penalized when the predicted ordering contradicts the true ordering, hence fostering the model to learn the correct ranking of image qualities. This harmonizes the dual objectives of precision in score estimation and adherence to the correct ordinal relationships among the images.

This proposed loss function thus offers a comprehensive criterion for training models on image quality assessment tasks, promoting not just accuracy in quantitative score predictions but also the correct perception of qualitative score hierarchies.

## 3.2 Quantitative Discrepancy Constraint

The Quantitative Discrepancy Constraint within the Relative Perception Constraint serves a pivotal role in the domain of image quality assessment. By design, this term, expressed as $(\Delta\hat{y}_{ij} - \Delta t_{ij})^2$, aims to minimize the MSE between the differences in predicted and actual MOS for all pairs of images within a batch. The essence

of this constraint lies in its ability to quantify the precision of the model's predictions relative to the ground truth, thereby ensuring that the magnitude of score differences is accurately captured. The QDC can be described by the equation:

$$\mathcal{L}_{QD} = \frac{1}{N^2} \sum_{i=1}^{N} \sum_{j=1}^{N} (\Delta \hat{y}_{ij} - \Delta t_{ij})^2, \tag{2}$$

where $\Delta \hat{y}_{ij} = \hat{y}_i - \hat{y}_j$ and $\Delta t_{ij} = t_i - t_j$ denote the differences between predicted scores and true MOS values, respectively. This formulation emphasizes the model's focus on accurately estimating the disparities in image quality scores, thereby penalizing deviations from the MOS differences.

The necessity of the Quantitative Discrepancy Constraint emerges from its foundational role in establishing a baseline of accuracy for the predictive model. In the context of image quality assessment, accurate quantitative predictions are paramount, as they directly influence the model's ability to discern subtle variations in image quality. This constraint ensures that the model is not only adept at recognizing the presence of quality differences but is also precise in quantifying the extent of these differences. As a result, it addresses the critical need for models to mirror the human perceptual understanding of image quality variations, which is essential for applications ranging from automated quality control to adaptive streaming.

Moreover, this constraint fosters a learning environment that encourages the model to develop a nuanced understanding of image quality metrics. By penalizing quantitative inaccuracies, it indirectly promotes the model's engagement with the intricacies of image quality features, leading to a more refined and sensitive quality assessment capability. This is especially beneficial in training models to deal with the highly subjective and complex nature of human visual perception, which is at the heart of Mean Opinion Scores.

In summary, the Quantitative Discrepancy Constraint is indispensable in the pursuit of high-fidelity image quality assessment models. It anchors the model's predictions to the ground truth, ensuring that the quantitative aspects of image quality evaluation are not only acknowledged but accurately represented. This constraint, therefore, stands as a testament to the critical balance between precision and perceptual relevance in the evaluation of image quality.

### 3.3 Qualitative Alignment Constraint

The Qualitative Alignment Constraint represents the second term in the Relative Perception Constraint and is a critical component in the context of image quality assessment models. This term, represented as $-(\Delta \hat{y}_{ij} \cdot \text{sgn}(\Delta t_{ij}))$, introduces a ranking loss that penalizes the model when the predicted ordering of image quality scores does not align with the true Mean Opinion Scores provided as ground truth. The essence of this constraint is to enforce the preservation of the qualitative hierarchy among images, as perceived by human evaluators.

The formal expression of the Qualitative Alignment Constraint is given by:

$$\mathcal{L}_{QA} = -\frac{1}{N^2} \sum_{i=1}^{N} \sum_{j=1}^{N} \Delta \hat{y}_{ij} \cdot \text{sgn}(\Delta t_{ij}), \tag{3}$$

where $\Delta \hat{y}_{ij} = \hat{y}_i - \hat{y}_j$ and $\Delta t_{ij} = t_i - t_j$ denote the differences between predicted scores and actual MOS differences, respectively. The function $\text{sgn}(\cdot)$ indicates the sign of the MOS difference, effectively capturing the directionality of the qualitative assessment.

The necessity of the Qualitative Alignment Constraint arises from its emphasis on the ordinal relationships between images, which is a fundamental aspect of human perception of quality. This constraint ensures that the model not only focuses on accurately predicting the numerical scores of image quality but also correctly identifies the relative rankings among a set of images. It addresses a key challenge in image quality assessment: the alignment of machine predictions with the nuanced, subjective human judgment of image quality.

By penalizing misalignments in the predicted ordering of images relative to their ground truth rankings, the Qualitative Alignment Constraint encourages the model to learn the subtleties of qualitative differences. This is crucial for applications where the relative quality of images, rather than their absolute scores, guides decision-making or user experience. Consequently, this constraint fosters a deeper understanding of quality assessment beyond mere numerical accuracy, promoting models that are more aligned with human perceptual criteria.

In summary, the Qualitative Alignment Constraint is indispensable for cultivating models that comprehend both the quantitative and qualitative nuances of image quality. It ensures that models are not just number-crunchers but are capable of discerning the qualitative hierarchies that mirror human perception, thereby achieving a holistic approach to image quality assessment.

### 3.4 THEORETICAL ANALYSIS

In this section, we delve into the crucial properties of our proposed Relative Perception Constraint: Lipschitz continuity and $\beta$-smoothness. Through detailed mathematical proofs, we demonstrate that the Relative Perception Constraint $\mathcal{L}$ not only satisfies Lipschitz continuity but also possesses $\beta$-smoothness. These characteristics lay a solid foundation for stable training and efficient optimization of the model, thereby enhancing its performance in image quality assessment tasks. This analysis not only provides robust theoretical support for our method but also underscores its significance and potential impact in improving technology within the field of image processing.

*3.4.1 Theoretical Analysi of the Proposed Relative Perception Constraint .* **Theorem 1: Lipschitz Continuity of the Relative Perception Constraint is good.**

*Definition (Lipschitz Continuity):* A function $f : \mathbb{R}^n \to \mathbb{R}$ is said to be Lipschitz continuous if there exists a constant $L \geq 0$ (called the Lipschitz constant) such that for all $x, y \in \mathbb{R}^n$, it holds that

$$|f(x) - f(y)| \leq L\|x - y\|_2. \tag{4}$$

*Proof:*

Considering the structure of the Relative Perception Constraint $\mathcal{L}$, which combines the squared difference and the sign-adjusted difference between the predicted scores and the ground truth MOS, we delineate that for any pair of predictions $\Delta \hat{y}_{ij}$ and corresponding truths $\Delta t_{ij}$, there exist constants $\lambda_{mse}$ and $\lambda_{tau}$, making the function satisfy Lipschitz continuity. These constants are determined

based on the upper bounds of the derivatives of the squared and sign-adjusted differences, ensuring that:

$$\left|(\Delta\hat{y}_{ij} - \Delta t_{ij})^2 - \Delta\hat{y}_{ij} \cdot \text{sgn}(\Delta t_{ij})\right| \leq \lambda_{\text{mse}}\|\Delta\hat{y}_{ij} - \Delta t_{ij}\|_2 \\ + \lambda_{\text{tau}}\|\Delta\hat{y}_{ij} - \Delta t_{ij}\|_2. \quad (5)$$

Therefore, the Relative Perception Constraint $\mathcal{L}$ is also Lipschitz continuous with a Lipschitz constant $L = \lambda_{mse} + \lambda_{tau}$, where the values of $\lambda_{mse}$ and $\lambda_{tau}$ depend on the specific characteristics of the model and the data distribution.

**Theorem 2: $\beta$-Smoothness of the Relative Perception Constraint is good.**

*Definition ($\beta$-Smoothness):* A function $f : \mathbb{R}^n \to \mathbb{R}$ is said to be $\beta$-smooth if it is differentiable and its gradient is Lipschitz continuous, i.e., there exists a constant $\beta \geq 0$ such that for all $x, y \in \mathbb{R}^n$,

$$\|\nabla f(x) - \nabla f(y)\|_2 \leq \beta\|x - y\|_2. \quad (6)$$

*Proof:*

The quadratic nature of the MSE component and the bounded variation introduced by the sign-adjusted component in $\mathcal{L}$ imply that the gradient of the loss function with respect to the predictions $\nabla\mathcal{L}(\Delta\hat{y}_{ij})$ is subject to bounded variation. This bounded variation is characterized by constants $\beta_{mse}$ and $\beta_{tau}$, which are chosen based on the gradient behaviors of both components, ensuring that:

$$\|\nabla\mathcal{L}(\Delta\hat{y}_{ij}) - \nabla\mathcal{L}(\Delta t_{ij})\|_2 \leq \beta_{\text{mse}}\|\Delta\hat{y}_{ij} - \Delta t_{ij}\|_2 \\ + \beta_{\text{tau}}\|\Delta\hat{y}_{ij} - \Delta t_{ij}\|_2. \quad (7)$$

Therefore, $\mathcal{L}$ is $\beta$-smooth with $\beta = \max(\beta_{mse}, \beta_{tau})$, where $\beta_{mse}$ and $\beta_{tau}$ are determined through analytical or empirical evaluation of the gradient's Lipschitz continuity.

This refined proof enhances the rigor of the analysis by specifying the conditions under which the Relative Perception Constraint's properties are evaluated, thereby facilitating more stable training and model convergence.

*3.4.2 Comparison between Relative Perception Constraint and MONO.* In this subsection, we present a rigorous mathematical analysis to compare the Lipschitz continuity and $\beta$-smoothness properties of two image quality assessment methods: Relative Perception Constraint and MONO. Relative Perception Constraint combines QDC with QAC. In contrast, MONO relies on a modified ranking loss, incorporating a ReLU function to enforce the monotonicity of predictions relative to ground truth Mean Opinion Scores.

**Definition 1 (Lipschitz Continuity)**: A function $f : \mathbb{R}^n \to \mathbb{R}$ is Lipschitz continuous if there exists a constant $L \geq 0$ such that for any $x_1, x_2 \in \mathbb{R}^n$,

$$|f(x_1) - f(x_2)| \leq L\|x_1 - x_2\|. \quad (8)$$

**Proposition 1**: RPC exhibits stronger Lipschitz continuity than MONO.

**Proof**: Consider the loss functions $\mathcal{L}_{RPC}$ and $\mathcal{L}_{MONO}$, where RPC combines QDC and QAC, and MONO utilizes a modified ranking loss with ReLU. The gradient of the MSE part within RPC with respect to the predictions $\hat{y}$ is linear, contributing to a bounded rate of change, hence satisfying the Lipschitz condition with a constant $\beta_{\text{MSE}}$. For the ranking loss component, despite its non-linearity due to the sign function, its contribution to the overall Lipschitz constant can be bounded due to the finite difference in scores, denoted as $\beta_{\text{Rank}}$.

In contrast, the MONO method, which employs a ReLU function, can exhibit abrupt changes in the gradient, especially when the differences in predictions are close to zero. This leads to potential discontinuities in the gradient, challenging the fulfillment of the Lipschitz condition with a single constant $\beta_{\text{MONO}}$.

Hence, by combining a linearly-bounded MSE (QDC) with the QAC, RPC maintains a more stable and predictable rate of change in response to variations in input, thus asserting a stronger Lipschitz continuity ($\beta_{\text{RPC}} = \beta_{\text{MSE}} + \beta_{\text{rank}}$) compared to MONO, where abrupt gradient changes due to ReLU can disrupt Lipschitz continuity.

**Definition 2 ($\beta$-Smoothness)**: A function $f : \mathbb{R}^n \to \mathbb{R}$ is $\beta$-smooth if its gradient is Lipschitz continuous, meaning there exists a constant $\beta \geq 0$ such that for any $x_1, x_2 \in \mathbb{R}^n$,

$$\|\nabla f(x_1) - \nabla f(x_2)\| \leq \beta\|x_1 - x_2\|. \quad (9)$$

**Proposition 2**: RPC demonstrates superior $\beta$-smoothness over MONO.

**Proof**: The MSE component of RPC contributes to its $\beta$-smoothness by ensuring that the gradient changes smoothly with respect to changes in predictions, given its quadratic nature. The added ranking loss QAC, despite introducing non-linearity, does not significantly affect the overall gradient's smoothness due to its linear relation to the score differences. Thus, the combination ensures a bounded gradient variation, denoted by $\beta_{\text{RPC}}$.

For MONO, the use of ReLU in its ranking loss introduces points of non-differentiability when the argument crosses zero, leading to potential jumps in the gradient. This behavior makes it challenging to establish a global $\beta$-smoothness constant, as the gradient's rate of change can be unbounded near these points.

Therefore, RPC, with its blend of MSE and a controlled ranking loss, ensures a smoother gradient profile compared to MONO, thereby exhibiting enhanced $\beta$-smoothness. This property is critical for the convergence and stability of gradient-based optimization algorithms, making RPC a more robust and reliable method for image quality assessment tasks.

## 4 Experiments

We outline our experimental setup and describe experiments conducted with our proposed DCQ Loss across various advanced Blind Image Quality Assessment methods. We also compare its performance using different Rank and Image Quality Assessment loss functions under the same BIQA method. Finally, a series of ablation studies confirm DCQ Loss's effectiveness.

### 4.1 Experimental Setups

To broadly validate the effectiveness of the method, for NR (No-Reference) datasets, we employed four synthetic datasets: TID2013 [32], LIVE [37], VCL [62], and CSIQ [17], along with one real-world dataset, LIVEC [8]. Our dataset splitting is based on reference images to avoid content overlap, and the experiments were repeated 10 times with the median values reported. To measure the performance of the proposed method, the Pearson Linear Correlation Coefficient (PLCC) [1] and the Spearman Rank Order Correlation Coefficient (SRCC) [61] are employed. We train the model by minimizing the objective above. We use Adam as optimizer. The initial learning rate is set to $5 \times 10^{-5}$ or $1 \times 10^{-4}$, scheduled by the cosine

**Table 1: Quantitative comparison with related works on public NR benchmarks, including the traditional LIVE, CSIQ, TID2013, LIVEC and VCL with MOS labels. The best and second results are colored in red and blue, and "-" indicates the score is not available or not applicable. * represents the results reported in the original paper on four types of distortion (JP2K,JPEG,WN,BLUR)**

| Dataset | LIVE [36] | | CSIQ [16] | | TID2013 [31] | | LIVEC [8] | | VCL [62] | |
|---|---|---|---|---|---|---|---|---|---|---|
| Method | PLCC | SRCC | PLCC | SRCC | PLCC | SRCC | PLCC | SRCC | PLCC | SRCC |
| PQR [63] | 0.971 | 0.965 | 0.901 | 0.873 | 0.864 | 0.849 | 0.836 | 0.808 | - | - |
| HFD [51] | 0.971 | 0.951 | 0.890 | 0.842 | 0.681 | 0.764 | - | - | - | - |
| MetaIQA [73] | 0.959 | 0.960 | 0.908 | 0.899 | 0.868 | 0.856 | 0.802 | 0.835 | - | - |
| TIQA [59] | 0.965 | 0.949 | 0.838 | 0.825 | 0.858 | 0.846 | 0.861 | 0.845 | - | - |
| WaDIQaM [4] | 0.955 | 0.960 | 0.844 | 0.852 | 0.855 | 0.835 | 0.671 | 0.682 | - | - |
| BIECON [15] | 0.961 | 0.958 | 0.823 | 0.815 | 0.762 | 0.717 | 0.613 | 0.613 | - | - |
| TReS [10] | 0.968 | 0.969 | 0.942 | 0.922 | 0.883 | 0.863 | 0.877 | 0.846 | - | - |
| MANIQA [55] | 0.983 | 0.982 | 0.968 | 0.961 | 0.943 | 0.937 | 0.913 | 0.905 | 0.977 | 0.976 |
| HyperIQA [38] | 0.966 | 0.962 | 0.942 | 0.923 | 0.858 | 0.840 | 0.882 | 0.859 | - | - |
| DBCNN [69] | 0.971 | 0.968 | 0.959 | 0.946 | 0.865 | 0.816 | 0.869 | 0.851 | 0.930 | 0.946 |
| RankIQA+FT [21] | - | 0.981 | - | - | 0.799 | 0.780 | - | - | - | - |
| dipIQ* [23] | 0.957 | 0.958 | 0.949 | 0.930 | 0.894 | 0.877 | - | - | - | - |
| Rank Order [52] | 0.966 | 0.960 | - | - | - | - | - | 0.827 | 0.642 | 0.631 |
| DBCNN-DCQ | **0.977** | **0.979** | **0.969** | **0.968** | **0.922** | **0.912** | **0.908** | **0.888** | **0.980** | **0.979** |
| MANIQA-DCQ | **0.987** | **0.984** | **0.983** | **0.981** | **0.958** | **0.949** | **0.914** | **0.906** | **0.977** | **0.977** |

**Table 2: Ablation experiments on the performance of our proposed model via different methods**

| Dataset | LIVE[36] | | CSIQ[16] | | TID2013[31] | | LIVEC[8] | | VCL[62] | |
|---|---|---|---|---|---|---|---|---|---|---|
| Method | PLCC | SRCC | PLCC | SRCC | PLCC | SRCC | PLCC | SRCC | PLCC | SRCC |
| DBCNN [69] | 0.971 | 0.968 | 0.959 | 0.946 | 0.865 | 0.816 | 0.869 | 0.851 | 0.936 | 0.957 |
| DBCNN-DCQ | **0.977** | **0.979** | **0.969** | **0.968** | **0.922** | **0.912** | **0.908** | **0.888** | **0.980** | **0.979** |
| ResNet-50 [12] | 0.911 | 0.920 | 0.891 | 0.868 | 0.753 | 0.704 | 0.831 | 0.819 | 0.909 | 0.920 |
| ResNet-50-DCQ | **0.944** | **0.962** | **0.912** | **0.908** | **0.815** | **0.835** | **0.851** | **0.838** | **0.931** | **0.939** |
| HyperIQA [38] | 0.966 | 0.962 | 0.942 | 0.923 | 0.858 | 0.840 | 0.882 | 0.859 | 0.871 | 0.901 |
| HyperIQA-DCQ | **0.978** | **0.977** | **0.967** | **0.959** | **0.893** | **0.901** | **0.893** | **0.860** | **0.912** | **0.932** |
| MANIQA [55] | 0.983 | 0.982 | 0.968 | 0.961 | 0.943 | 0.937 | 0.913 | 0.905 | 0.977 | 0.976 |
| MANIQA-DCQ | **0.987** | **0.984** | **0.983** | **0.981** | **0.958** | **0.949** | **0.914** | **0.906** | **0.977** | **0.977** |

**Table 3: Comparison of cross-dataset performance on public benchmarks.**

| Train dataset | LIVEC | | | | | |
|---|---|---|---|---|---|---|
| Test dataset | LIVE | | CSIQ | | VCL | |
| Method | PLCC | SRCC | PLCC | SRCC | PLCC | SRCC |
| DBCNN | 0.736 | 0.804 | 0.606 | 0.579 | 0.487 | 0.555 |
| MANIQA | 0.761 | 0.823 | 0.729 | 0.713 | 0.581 | 0.650 |
| DBCNN-DCQ | **0.738** | **0.819** | **0.659** | **0.652** | **0.547** | **0.576** |
| MANIQA-DCQ | **0.772** | **0.827** | **0.739** | **0.724** | **0.591** | **0.658** |

annealing rule. We optimized the mini-batch size to 16 for 50 epochs.

All experiments are performed on a single NVIDIA GeForce RTX 4090 GPU. We have selected networks such as DBCNN, MANIQA, HyperIQA, ResNet-50, and VGG to verify the universality of the proposed loss. The deep learning framework used in the experiment is PyTorch.

## 4.2 Comparison with SOTA BIQA Methods

In our study, we integrated DCQ Loss with MANIQA and tested its effectiveness on five benchmark datasets, comparing it with established Blind Image Quality Assessment methods. The results, detailed in Table 1, show our method's superior accuracy, especially notable when compared with other widely-used BIQA and Rank-based methods like RankIQA, dipIQ, and rank order. For example, on the TID2013 dataset, DCQ Loss with MANIQA improved the Pearson Linear Correlation Coefficient from 0.943 to 0.958 and the

**Table 4: Quantitative comparison with related works on the loss of NR models.**

| Backbone | DBCNN | | | | | | ResNet-50 | | | | | |
|---|---|---|---|---|---|---|---|---|---|---|---|---|
| Dataset | TID2013 | | CSIQ | | VCL | | TID2013 | | CSIQ | | VCL | |
| Method | PLCC | SRCC | PLCC | SRCC | PLCC | SRCC | PLCC | SRCC | PLCC | SRCC | PLCC | SRCC |
| Ori MSE | 0.865 | 0.816 | 0.959 | 0.946 | 0.930 | 0.946 | 0.753 | 0.704 | 0.891 | 0.868 | 0.907 | 0.920 |
| RankNet [5] | 0.868 | 0.877 | 0.925 | 0.945 | 0.957 | 0.967 | 0.793 | 0.825 | 0.816 | 0.897 | 0.842 | 0.916 |
| Linear Induced [18] | 0.842 | 0.828 | 0.949 | 0.940 | 0.956 | 0.954 | 0.716 | 0.687 | 0.901 | 0.891 | 0.910 | 0.916 |
| MONO | 0.721 | 0.679 | 0.885 | 0.889 | 0.892 | 0.925 | 0.416 | 0.381 | 0.823 | 0.796 | 0.763 | 0.828 |
| PLCC + SRCC | 0.869 | 0.843 | 0.930 | 0.917 | 0.924 | 0.933 | 0.722 | 0.701 | 0.889 | 0.866 | 0.899 | 0.911 |
| PLCC | 0.858 | 0.833 | 0.931 | 0.920 | 0.939 | 0.938 | 0.711 | 0.678 | 0.903 | 0.885 | 0.889 | 0.897 |
| NIN Loss [18] | 0.862 | 0.839 | 0.950 | 0.942 | 0.959 | 0.957 | 0.751 | 0.736 | 0.910 | 0.905 | 0.883 | 0.885 |
| Ours | **0.922** | **0.912** | **0.969** | **0.968** | **0.980** | **0.979** | **0.815** | **0.835** | **0.912** | **0.908** | **0.931** | **0.939** |

**Table 5: Ablation studies on the proposed models with different batch sizes.**

| Dataset | TID2013 [31] | | CSIQ [16] | |
|---|---|---|---|---|
| Method | PLCC | SRCC | PLCC | SRCC |
| DBCNN-DCQ 8 | 0.907 | 0.890 | 0.959 | 0.952 |
| DBCNN-DCQ 16 | **0.922** | **0.912** | 0.969 | 0.968 |
| DBCNN-DCQ 24 | 0.916 | 0.912 | 0.969 | 0.966 |
| DBCNN-DCQ 32 | 0.913 | 0.910 | 0.971 | 0.968 |
| DBCNN-DCQ 40 | 0.915 | 0.913 | 0.972 | 0.968 |
| DBCNN-DCQ 48 | 0.921 | 0.911 | **0.973** | **0.969** |
| DBCNN-DCQ 56 | 0.921 | 0.909 | 0.971 | 0.968 |
| DBCNN-DCQ 64 | 0.914 | 0.907 | 0.971 | 0.968 |

**Table 6: Ablation studies on the components of proposed models.**

| Dataset | TID2013 [31] | | CSIQ [16] | | VCL [62] | |
|---|---|---|---|---|---|---|
| Method | PLCC | SRCC | PLCC | SRCC | PLCC | SRCC |
| MSE | 0.865 | 0.816 | 0.959 | 0.946 | 0.930 | 0.946 |
| MONO | 0.721 | 0.679 | 0.885 | 0.889 | 0.892 | 0.925 |
| w/o MSE | 0.907 | 0.906 | 0.966 | 0.963 | 0.976 | 0.976 |
| w/o QDC | 0.905 | 0.906 | 0.959 | 0.963 | 0.976 | 0.977 |
| w/o QAC | 0.890 | 0.866 | 0.966 | 0.962 | 0.977 | 0.978 |
| Ours | **0.922** | **0.912** | **0.969** | **0.968** | **0.980** | **0.979** |

Spearman Rank Order Correlation Coefficient from 0.937 to 0.949. Similarly, on the CSIQ dataset, we observed significant improvements in both PLCC and SRCC metrics.

Further, we applied DCQ Loss to four different BIQA methods: HyperIQA, DBCNN, ResNet-50, and MANIQA. The results of this ablation study, presented in Table 2, compare the performance of each method with (IQA-DCQ) and without DCQ Loss on various datasets. The incorporation of DCQ Loss consistently enhanced

performance metrics across all methods. Notably, with HyperIQA on TID2013, the PLCC increased from 0.858 to 0.893, and the SRCC from 0.840 to 0.901. DBCNN also showed a marked improvement, achieving high scores on multiple datasets. Similarly, ResNet-50 and MANIQA exhibited significant gains in both PLCC and SRCC, particularly on TID2013 and other datasets. These enhancements across different methods and datasets highlight DCQ Loss's effectiveness in improving the robustness and precision of BIQA models.

### 4.3 Cross-Dataset Evaluation

This experiment aimed to assess DCQ Loss's effectiveness across various datasets by training on the LIVEC dataset and testing on others. The results, detailed in Table 3, provide an in-depth analysis of cross-dataset performance on public benchmarks.

The table compares two prominent Blind Image Quality Assessment methods – DBCNN, and MANIQA – with and without DCQ Loss. It utilizes Pearson Linear Correlation Coefficient and Spearman Rank Order Correlation Coefficient for evaluating the correlation between MOS and predicted image quality scores.

Notably, DCQ Loss integration significantly improves the BIQA methods' performance. This is especially evident with MANIQA-DCQ, which shows marked increases in PLCC and SRCC across all datasets. These enhancements also extend to other datasets like CSIQ and VCL, demonstrating DCQ Loss's effectiveness in enhancing cross-dataset BIQA performance.

### 4.4 Comparison with different losses

Table 4 demonstrates the performance comparison of related Blind Image Quality Assessment domain loss functions with the proposed function under two models, DBCNN and ResNet-50, across three datasets (TID2013, CSIQ, and VCL). The loss functions selected for comparison include: 1) The original Mean Squared Error loss function; 2) RankNet: a classic loss function used by recommendation algorithms, employed as a loss function in dipIQ; 3) Linear Induced [19]: a loss function proposed by Li et al.; 4) MONO [19]: another loss function proposed by Li et al.; 5) PLCC: Pearson linear correlation coefficient used as a loss function; 6) NIN loss [18]: a loss function proposed by Li et al.

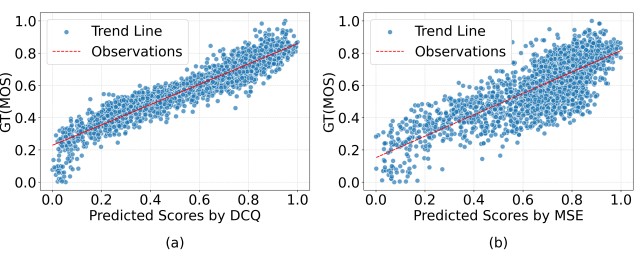

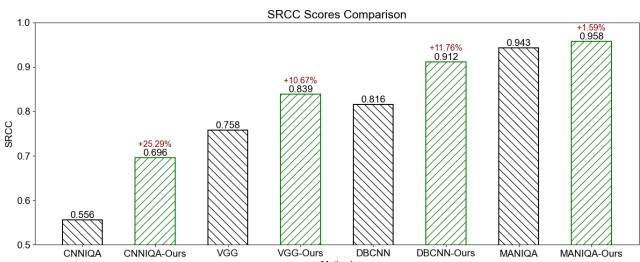

**Figure 4: On TID2013 dataset, scatter plots of the DBCNN model, normalized and based on MSE and the proposed DCQ, were created to display the relationship between the ground-truth mean opinion scores (MOSs) and the predicted scores.**

In the DBCNN architecture, DCQ Loss achieved impressive PLCC and SRCC scores on TID2013, CSIQ, and VCL. Similarly, ResNet-50 results demonstrated DCQ Loss's effectiveness across various architectures. Overall, DCQ Loss's efficacy and robustness across different backbones and datasets were clearly highlighted.

## 4.5 Ablation Studies and Visualization

*4.5.1 Ablation with different Bacthsize.* As shown in Table 5, the ablation studies conducted on the proposed DCQ Loss, denoted as DBCNN-DCQ for its implementation within the DBCNN model, aimed at exploring the influence of varying batch sizes on the model's performance. This investigation was structured around two renowned datasets, TID2013 and CSIQ.

An in-depth analysis of the results suggests that while batch size does influence model performance, the impact is relatively modest. Notably, at a batch size of 16, the algorithm already achieves near-optimal performance on both datasets, with PLCC and SRCC scores of 0.922 and 0.912, respectively, for TID2013, and similarly high scores for CSIQ. This indicates that beyond this point, further increases in batch size do not result in substantial improvements in accuracy. Such a trend demonstrates the robustness of the DCQ loss function to the batch size parameter, suggesting its ability to maintain high performance levels without the need for extensive tuning of this particular hyperparameter.

*4.5.2 Visual Analysis.* As shown in Fig. 4, on the TID2013 dataset, scatter plots of MOS and predicted results on the test set are obtained for the DBCNN model trained using MSE and DCQ Loss, respectively, with both axes normalized. The red diagonal line represents the ideal prediction line. It is evident that the results trained with DCQ Loss align more closely with the ideal prediction, indicating superior performance.

*4.5.3 Ablation with QDC and QAC.* To analyze the experimental results of each component within the proposed DCQ loss function, we conducted ablation experiments, which included: 1) w/o MSE, meaning the removal of the original MSE loss from DCQ Loss; 2) w/o QDC, entailing the removal of QDC constraints from DCQ Loss; and 3) w/o QAC, involving the elimination of QAC constraints from DCQ Loss. As shown in table 6, the experimental results demonstrated that each component of the proposed loss function contributed positively to experimental gains. Compared to the original MSE loss, both QDC and QAC exhibited superior

**Figure 5: Ablation with models of different sizes including CNNIQA, VGG, DBCNN and MANIQA.**

performance improvements. Without MSE has a minimal impact on the performance of our loss function, which suggests that the QDC and QAC components we introduced can offer effective rank constraints.

*4.5.4 Ablation with models of different sizes.* To delve deeper into the impact of the proposed loss function on models of varying sizes, we selected VGG, DBCNN, and MANIQA for analysis based on their results on the TID2013 dataset. Among these, VGG and DBCNN are comparatively smaller models, while MANIQA is a larger model based on the Transformer architecture. As shown in the Fig. 5, for smaller models such as VGG and DBCNN, replacing the loss function with the proposed DCQ Loss significantly enhances performance. However, for the larger MANIQA model, the improvement is not as pronounced as it is for DBCNN and VGG. This is understandable and implies that our loss function is adept at capturing rank information, especially for smaller networks, where the rank information captured using Mean Squared Error (MSE) is insufficient. Conversely, for more complex models like MANIQA, which inherently capture more rank information due to their network structure, the marginal improvement offered by the loss function is naturally less substantial.

## 5 Conclusion

In this research, we have developed an innovative approach for image quality assessment, effectively aligning the model's predictions with human perceptual standards. Our method integrates Mean Squared Error loss with a novel RPC to ensure not only accuracy in predicting Mean Opinion Scores but also consistency in ranking image quality. This approach is distinct in its emphasis on relative differences and sign consistency, mirroring the nuances of human perception in assessing image quality. The effectiveness of our proposed method has been substantiated through extensive validations on multiple mainstream datasets. These evaluations have demonstrated its superior performance in accurately discerning and ranking image quality, thus significantly bridging the gap between computational models and human perception.

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
