# OpenReview forum: "Dual-Criterion Quality Loss for Blind Image Quality Assessment"
_acmmm.org/ACMMM/2024/Conference — MM2024 Oral_

### Official Review · Reviewer_2fSx · 2024-05-09

**Rating:** 4
**Confidence:** 4

**Summary:**

This paper proposes a loss function called dual-criterion qualiity loss for image quality assessment (IQA), the novel loss function consists of root mean square error (MSE) and relative perception constraint (RPC). PRC takes into account both the mean square error and the ordinal relationship of the differences between the predicted score differences and the mean opinion score (MOS) differences. The extensive experiments are conducted to verify the effectiveness of this approach.

**Strengths:**

The motivation of this paper is very clear and the logic is clear, and the importance of the proposed loss function for IQA is analyzed through a large number of theories. In addition, DCQ loss significantly improves existing IQA methods. Reliable theoretical analysis and experimental results are the specialty of this paper

**Limitations:**

I still have some problems with the proposed loss function, and there are still many small problems in the paper, which are listed below:

Main issues:

In fact, DCQ loss is a loss combination of MSE+RPC(QAC+QDC), where RPC takes into account both the mean square error and the ordinal relationship of the differences between the predicted score differences and the MOS differences. So, who has the advantage between PRC and (PLCC+SRCC) loss combination? Because the article also mentioned the article [43]. Furthermore, the proposed loss function is usually only compared with a single loss function rather than a combination of multiple loss functions, which I think is relatively unfair. All in all, I want to know how RPC compares with (PLCC+SRCC) loss.

Minor issue:
(1) It is strange that SVR appears in Figure 1, because SVR is a regression module.
(2) Line 230, " Zhang et al. utilized discrete ranking ....", references are missing here.
(3) Lines 236-238, ". Ma et al. [24, 26] extracted binary ranking information during the training process through FR-IQA (Full Reference Image Quality Assessment) methods." On the one hand, there is a problem with abbreviations. The full name of the noun should be outside the brackets, while the abbreviation should be within the brackets. On the other hand, literature [24] and [26] are blind IQA methods, why are they described as FR-IQA here?
(4) I did not find a description of Table 3 in the text.
(5) Section 3 should provide the complete formula for the DCQ loss function
(6) Section 4.1. (i) Lines 567-569, "Our dataset splitting is based on reference images to avoid content overlap, and the experiments were repeated multiple times with the median values reported." Please describe specifically how many times it has been conducted? (ii) Line 572, "......(SROCC) [55] are employed." The abbreviation SROCC is used here, but elsewhere and in experimental tables the abbreviation SRCC is used. (iii) Line 574, "......is set to 5×10−5 or 1×10−4", why are two initial learning rates set?
(6) The title of Table 2 should be modified. This is an ablation experiment of the proposed loss function on different IQA methods, rather than different backbone networks. Because Hyper-IQA and ResNet-50 both use ResNet-50 as the backbone network.
(7) Lines 683-685, "HyperIQA, DBCNN, ResNet-50, and MANIQA. The results of this ablation study, presented in Table 2, compare the performance of each method with and without DCQ Loss (-DCQ) on various datasets." Descriptive error. "with DCQ loss" should be denoted as "IQA-DCQ"  rather than "without DCQ Loss".
(8) The title of Table 6 should be wrong.
(9) Please check the full paper for abbreviations. Abbreviations should be defined the first time they occur, and abbreviations throughout the paper are very confusing and are often defined repeatedly.

[43] Haoning Wu, Chaofeng Chen, Liang Liao, Jingwen Hou, Wenxiu Sun, Qiong Yan, Jinwei Gu, and Weisi Lin. 2023. Neighbourhood representative sampling for efficient end-to-end video quality assessment. IEEE Transactions on Pattern Analysis and Machine Intelligence (2023).

**Suitability:**

3

---

### Official Review · Reviewer_J6VH · 2024-05-22

**Rating:** 5
**Confidence:** 3

**Summary:**

This paper introduced a dual-criterion quality (DCQ) loss for blind image quality assessment. The DCQ which contains a relative perception constraint (RPC) and a quantitative discrepancy constraint (QDC), provides a more straightforward and efficient approach. The authors prove the efficiency of DCQ loss in both theoretical and experimental perspectives.

**Strengths:**

The innovation of this article is novel, with clear motivation and rich theoretical and experimental evidence. Furthermore, the writing is flunet.

**Limitations:**

1. The author should strength the contribution in the introduction part.
2. I notice the loss definition in Eq.(1), which captures the numerical differences between prediction in the batch and itself. Will this operation affect the efficiency of DCQ loss?
3. In page 2, line 230, a reference is missing.
4. Table 2 should be merged with Table 1.
5. The result of MONO loss in Table 4 is even lower than Ori MSE. As far as i know, there are relative_score, mapped_score, and aligned_score in Li et al.[1] The MONO loss is only used to supervise the aligned_score. And then, Li et al. used the mapped_score to calculate the PLCC and SROCC. Is the comparative experiment fair?

**Suitability:**

3

---

### Official Review · Reviewer_m87F · 2024-05-24

**Rating:** 4
**Confidence:** 3

**Summary:**

The paper proposes Dual-Criterion Quality Loss, a stable, efficient loss function aimed to the lack of information on image ranking in the Mean Squared Error (MSE) loss. Dual-Criterion Quality Loss combines MSE with Relative Perceptual Constraints (RPC). RPC includes Quantitative Difference Constraint (QDC) and Qualitative Alignment Constraint (QAC). Lipschitz continuity and β-smoothness are proved through rigorous mathematical proofs.

**Strengths:**

1. The motivation of this work is novelty.
2. The theoretical proof of the loss function is reliable.

**Limitations:**

1 Although the paper emphasizes the stability and effectiveness of the proposed methods, it does not mention computational efficiency, such as inference time of the model, resource consumption, etc., which is an important consideration for practical applications.
2 It is suggested to conclude the contributions of this work in the introduction, it may be done from both theoretical proof and experimental perspectives.
3. The QDC Loss has been validated on DBCNN (2019), HyperIQA (2020), and ResNet50 (2016), and the effectiveness of the DQC loss needs to be validated in the latest IQA methods (2022~2024).

**Suitability:**

3

---

### Official Review · Reviewer_EPdb · 2024-05-24

**Rating:** 3
**Confidence:** 2

**Summary:**

This paper addresses a problem of optimization for typical trained models used in quality assessment by introducing a novel loss function model.

**Strengths:**

Results show improvements against typical approaches.

**Limitations:**

The paper is wrote in a confusing way and it is difficult to read!

Quite limited contribution.


Remarks
The sentence in line 108 is controversial. The paper confuses MSE with subjective quality. Anyone that uses MSE for optimization intends to achieve the best possible representation of the original signal. The perceived quality is something different.

Labels in fig. 2 and scales are unacceptably small.

I consider eq 1 some how unbalanced because subtracts x to x^2. So different dimensions combined linearly appear in the expression, which is odd for me!
Is (1) differentiable? --i believe it is not, so how to have a loss function with it?

Lines 577, 578 add citations to the different models.

**Suitability:**

2

---

### Meta-Review · Area_Chair_1FnH · 2024-07-04

**Recommendation:** Accept (Oral)
**Confidence:** 5

**Metareview:**

The paper tackles the problem of blind (no-reference) image quality assessment using deep learning architectures. One of the issues that these algorithms face is the ground truth, which consist of quality scores provided by human participants in a subjective experiment. A single number (the mean observer score - MOS) is provided for the whole image (and for a video, in the case of a video quality assessment method). Naturally, for these algorithms an important parameter is the loss metric used by the algorithm. Often MSE or another quantitative metric of the distance between the two images (reference and test) is used. The authors focus on proposing a loss metric that improves the performance of the metric.

This is certainly a very important topic, as demonstrated in the results in Fig. 1. The paper is nicely written, with a clear explanation of the problem. The experimental validation is sufficient. Nevertheless, I agree with one of the reviewers the contribution is somewhat limited. Also, there are some issues with the proposed loss that were not answered by the authors.